# Cellulosic Ethanol Production Using Waste Wheat Stillage after Microwave-Assisted Hydrotropic Pretreatment

**DOI:** 10.3390/molecules27186097

**Published:** 2022-09-18

**Authors:** Grzegorz Kłosowski, Dawid Mikulski, Prashant Bhagwat, Santhosh Pillai

**Affiliations:** 1Department of Biotechnology, Faculty of Biological Sciences, Kazimierz Wielki University, ul. K. J. Poniatowskiego 12, 85-671 Bydgoszcz, Poland; 2Department of Biotechnology and Food Science, Faculty of Applied Sciences, Durban University of Technology, P.O. Box 1334, Durban 4000, South Africa

**Keywords:** microwave-assisted pretreatment, hydrotrope, distillery stillage, bioethanol

## Abstract

One of the key elements influencing the efficiency of cellulosic ethanol production is the effective pretreatment of lignocellulosic biomass. The aim of the study was to evaluate the effect of microwave-assisted pretreatment of wheat stillage in the presence of sodium cumene sulphonate (NaCS) hydrotrope used for the production of second-generation bioethanol. As a result of microwave pretreatment, the composition of the wheat stillage biomass changed significantly when compared with the raw material used, before treatment. Microwave-assisted pretreatment with NaCS effectively reduced the lignin content and hemicellulose, making cellulose the dominant component of biomass, which accounted for 42.91 ± 0.10%. In post pretreatment, changes in biomass composition were also visible on FTIR spectra. The peaks of functional groups and bonds characteristic of lignins (C–O vibration in the syringyl ring, asymmetric bending in CH_3_, and aromatic skeleton C–C stretching) decreased. The pretreatment of the analyzed lignocellulosic raw material with NaCS resulted in the complete conversion of glucose to ethanol after 48 h of the process, with yield (in relation to the theoretical one) of above 91%. The highest observed concentration of ethanol, 23.57 ± 0.10 g/L, indicated the high effectiveness of the method used for the pretreatment of wheat stillage that did not require additional nutrient supplementation.

## 1. Introduction

Limited resources of fossil fuels encourage the search for cheap and environmentally friendly methods of obtaining fuels from renewable sources, such as second-generation bioethanol. Large-scale production of ethanol from waste lignocellulosic raw materials does not pose a risk of increasing food and feed prices, as is the case for ethanol production from starchy raw materials [1]. From a technological point of view, the use of lignocellulose is much more difficult, therefore it is very important that the methods of producing cellulosic bioethanol are as efficient and environmentally friendly as possible. The highly efficient production of ethanol from plant biomass is conditioned by an effective pretreatment method, which reduces the degree of cellulose crystallinity, degrades hemicellulose and lignins, and thus facilitates the effective enzymatic decomposition of structural polysaccharides [2,3]. The effectiveness of the pretreatment can be assessed by analyzing the crystallinity of the biomass. An XRD analysis shows the proportion of crystalline substances in plant biomass, such as cellulose. The process of enzymatic hydrolysis is intensified when the crystallinity of cellulose is lowered [3]. In addition, the lignocellulose pretreatment process should not generate fermentation inhibitors or use chemicals that pollute the environment. A promising direction in the development of pretreatment methods is the use of environmentally safe and recoverable chemicals such as ionic liquids, deep eutectic solvents (DES), organosolv, or hydrotropes [4,5,6,7,8].

Hydrotropes are sodium and potassium salts of an alkyl group substituted benzoic and aryl sulfonic acids with an amphiphilic structure. Due to the presence of smaller hydrophobic fragments, hydrotropes do not form micelles [9]. The ability of hydrotropes to lower the surface tension and increase the solubility of hydrophobic compounds in aqueous solutions is used during the pretreatment of lignocellulose to delignify the raw material. Removal of lignins from plant biomass is crucial for efficient enzymatic hydrolysis of cellulose [3]. An additional advantage of using hydrotropes in the pretreatment of lignocellulosic biomass is the possibility of recovering lignins from the hydrotropic solution by diluting it with water. It is also possible to recover the hydrotrope from the aqueous solution by evaporating the water under reduced pressure and reusing it, thereby reducing the cost of raw materials and the impact on the environment [10,11]. Studies have shown that using hydrotropes at a concentration of 40% (w v^−1^) reduced the content of lignins by over 40% in forestry waste. However, elevated temperature and pressure are required for an effective delignification [12,13]. Microwave radiation is an effective and economically viable method of heating biomass to a precisely defined temperature in a short time. Moreover, microwaves used in the treatment of lignocellulosic biomass cause a partial degradation of hydrogen bonds as a result of the oscillating movement of dipoles and electromagnetic radiation [14]. Microwave radiation also intensifies the delignification process, shown in studies using the biomass of *Sesamum indicum* [15] and *Prosopis juliflora* [16]. The process of biomass delignification with the use of various solvents can be intensified with microwave radiation. The simultaneous use of DES and microwaves reduces the resistance of biomass to degradation, facilitates the process of enzymatic hydrolysis of cellulose, and promotes the formation of lignin nanoparticles [17]. The use of acids and bases in conjunction with exposure to microwave radiation (optimally adjusted to the type of biomass) also promotes the delignification process. The removal of lignins from the biomass of maize stems was achieved with an efficiency of nearly 92% within 5 min, which was significantly faster compared to conventional heating. Susceptibility to enzymatic hydrolysis with cellulases was also higher for microwaves pretreated biomass [18].

One of the many technical problems related to the production of cellulosic ethanol, in addition to the need for an effective pretreatment of lignocellulosic biomass without the use of substances harmful to the environment, is to obtain fermentation media that do not contain inhibitors limiting the efficiency of the process. The use of concentrated mineral acids during pretreatment at elevated temperatures leads to dehydration of sugars and the formation of compounds that adversely affect yeast metabolism [19]. Fermentation inhibitors include compounds such as furfural derived from xylose and 5-hydroxymethylfurfural (5-HMF) derived from glucose [20]. Therefore, in the cellulosic ethanol production technology, it is advisable to use pretreatment methods that prevent the generation of environmentally hazardous substances and provide a substrate with reduced lignin content, susceptible to further enzymatic hydrolysis. Differences in the composition and structure of lignocellulosic raw materials mean that developing economically viable methods for second-generation bioethanol production requires a good combination of effective pretreatment methods with an easily available waste source, lignocellulose. Distillery stillage, which is a waste of first-generation ethanol production from starch raw materials, is one of the by-products of the agri-food industry that can be successfully used as a raw material for the production of cellulosic ethanol [7].

The above-mentioned problems in the production of second-generation cellulosic ethanol became an impulse for research on the use of pretreated waste distillery stillage. Thus, the aim of the study was to develop a method for the effective use of wheat stillage biomass subjected to microwave hydrotropic treatment for the production of cellulosic ethanol. Waste wheat stillage was chosen for the study because it was readily available due to the large-scale production of first-generation ethanol. The pretreatment method was selected based on previous studies [12], in which the use of sodium cumene sulfonate (NaCS) together with exposure to microwave radiation ensured controlled barothermic parameters. Our previous research and studies of other authors confirmed the usefulness of NaCS as a reagent enabling the effective delignification of biomass. The use of NaCS for the pretreatment of lignocellulose also ensures a high degree of sugar conversion to ethanol, as fermentation inhibitors (5-HMF and furfural) are not formed [5,7]. It should be noted that the expected effects of using NaCS in the pretreatment of biomass were only achieved in conjunction with elevated pressure. The use of microwaves allows high pressure to be obtained in a short time, with the simultaneous control of parameters and the possibility of immediate termination of heating [21]. The advantage of the combined use of microwaves and a hydrotrope is the possibility of using the latter in a recirculation system (recovery of the hydrotrope after removing lignins by diluting the solution and then distilling off the water). Therefore, the proposed pretreatment method minimizes the environmental burden of the cellulosic bioethanol production process. The combination of this concept with the technological solution, including the simultaneous use of NaCS and microwaves for the pretreatment of wheat stillage is a novelty in the development of second-generation ethanol technology. 

## 2. Results and Discussion

### 2.1. Influence of NaCS Concentration during Microwave Pretreatment on Biomass Composition and Its Usefulness in the Alcoholic Fermentation Process

In the first stage of the research, the effect of NaCS concentration (0, 10, 20, and 40% *w*/*v*) used during the microwave pretreatment of wheat stillage biomass under constant process conditions (microwave generator power 300 W, exposure time 30 min, 0.81 MPa) on the biomass extractives, the amount of basic biomass components, that is cellulose, hemicellulose and lignins, and on the crystallinity index was assessed. Individual functional groups were also identified and quantified. As the concentration of NaCS increased (from 10 to 40% *w*/*v*), the amount of substance extracted (biomass extractives) during microwave pretreatment of wheat stillage biomass increased from 60.79 to 71.15% (Table 1). The regression coefficient (R^2^) for these values was 0.9969. The least amount of biomass extractives (48.57%) was found in the absence of the hydrotrope (0% *w*/*v* NaCS) (Table 1).

The observed effect is the result of an increase in the solubility of hydrophobic substances with an increase in NaCS concentration. A higher concentration of hydrotropes capable of lowering the surface tension increases the solubility of sparingly soluble substances in aqueous solutions. The microwave radiation used in the extraction process was reported to additionally enhance this effect [9,14]. A similar relationship was also found when hydrotropes were used for pretreating rice straw at 80 °C (0.05 MPa) for 4 h. With the increase of NaCS concentration from 10 to 30% *w*/*v*, the biomass extractives increased from ca. 15 to 22%. It is worth noting that the use of NaCS in this study during the microwave pretreatment of wheat stillage biomass allowed for a much greater (more than 3-fold) increase in biomass extractives compared to the results described in the cited publication, where rice straw was used [5]. When this pretreatment method was used in the previous study on corn stillage biomass as a source of lignocellulose, a similar relationship was observed; however, the maximum amount of the substance extracted using 20% *w*/*v* NaCS was then ca. 67% [12].

The analysis of the wheat stillage biomass (cellulose, hemicellulose, and lignins) after microwave treatment with increasing NaCS concentrations (from 0 to 40% *w*/*v*) revealed some regularities. Microwave pretreatment using various solvents significantly changed the composition of the biomass. The method used (40% *w*/*v* NaCS) significantly increased the percentage of cellulose (by 230%) and lignins (by 280%), and decreased the percentage of hemicellulose (by 95%) when compared with the biomass before pretreatment. The observed increase in the amount of cellulose and lignins is due to the fact that these components are not degraded during the microwave pretreatment, while the other components are extracted. The decrease in hemicellulose percentage is as a result of the decomposition of this compound at elevated temperatures. The cellulose percentage in biomass increased by ca. 15% DW, which is the result of using a constant amount of hydrotrope pretreated biomass to determine the cellulose percentage. The use of a constant weight of biomass to determine the cellulose percentage in combination with the extraction of components during hydrotropic pretreatment (such as hemicellulose and lignins) increased the observed cellulose concentration in biomass and is the result of the methodology adopted, which is also used by other authors [13,22]. Contrary to the cellulose percentage, the hemicellulose percentage in biomass decreased with increasing hydrotrope concentrations during microwave pretreatment. The hemicellulose percentage in the wheat stillage biomass after pretreatment with 40% *w*/*v* NaCS was lower by about 79% than in the biomass after the water pretreatment (0% *w*/*v* NaCS) (Table 1). A similar relationship was observed for lignins, but the percentage of lignins decreased by ca. 27% in biomass after the application of 40% *w*/*v* NaCS (Table 1). The components of the wheat stillage biomass that may not be extracted as a result of microwave hydrotropic treatment are fats, nitrogenous substances, phenolic compounds and minerals [23]. Hydrotropes lower the surface tension, thus increasing the solubility of hydrophobic compounds, including lignins. Increasing the solubility of phenolic compounds in the solution during the pretreatment process enables delignification, which is additionally intensified by microwave radiation [24,25,26]. The observed relationship, indicating the key impact of hydrotrope concentration on the degree of delignification of lignocellulosic biomass, was also observed in the studies of other authors. The reduction in lignin percentage in forestry waste, sugarcane bagasse, maize distillery stillage and wheat straw increased with increasing hydrotrope concentrations. The effectiveness of delignification of forestry waste with benzene sulphonic acid (in increasing concentrations) was up to 60% [13]. While the pretreatment of sugarcane bagasse with increasing concentrations of sodium xylene sulfonate (SXS) reduced the concentration of lignins by up to 80%, but the pretreatment took up to 300 min [27]. The delignification efficiency of up to 43% was found during the pretreatment of maize stillage with NaCS in increasing concentrations [12] and SXS used in the pretreatment of wheat straw [28]. In the present study, the delignification level was only about 27%, but it was achieved with over 70% loss of biomass components as a result of extraction at the pretreatment stage. The use of a constant weight of biomass for the determination of lignocellulosic components underestimates the calculated effectiveness of the delignification process due to the loss of biomass components at the pretreatment stage. The advantages of combining microwave distillery stillage treatment with another pretreatment method were also demonstrated by other authors. Simultaneous application of microwaves and biological treatment of *Phanerochaete chrysosporium* resulted in the efficiency of hydrolysis similar to that presented in this study. The use of white-rot fungi together with microwave radiation in the pretreatment of stillage biomass reduced the lignin content by 33%, which confirmed the effectiveness of microwaves as a factor intensifying the delignification process [29]. The possibility of reducing the content of lignins and hemicellulose in biomass at a level similar to that presented in this study was also observed by authors who used microwave radiation for the pretreatment of corn stalks. The use of water with microwave radiation enabled the extraction of only water-soluble components and increased the percentage of lignins in the biomass [30].

The relative biomass crystallinity index (CrI) determined by means of the XRD technique is an important parameter characterizing biomass samples after pretreatment [31,32]. As a result of microwave pretreatment, the relative crystallinity index increased from 70.15% for the pretreatment with water to 78.80% for 40% *w*/*v* NaCS (Figure 1, Table 1). Furthermore, XRD analyses showed an increase in the biomass relative crystallinity index for pretreatment methods where partial removal of hemicellulose and lignins was observed. Removal of lignins from biomass and partial hydrolysis of hemicellulose during microwave pretreatment with NaCS reduces the amount of amorphous substances in the biomass and increases the content of the crystalline fraction [33]. The increase in the relative crystallinity of biomass was also observed in the studies of other authors. For example, the crystallinity index of wheat straw increased from 68 to 71% as a result of pretreatment with ionic liquid [34]. Also, microwave or ultrasonic pretreatment increased the crystallinity of agave biomass from 70 to 73% [35]. Thus, the XRD analyses confirmed the effectiveness of the applied microwave hydrotropic treatment in the delignification of wheat stillage biomass.

The results of XRD were also supported by FTIR analyses (Figure 2). The peaks in the FTIR spectra in areas characteristic of functional groups and bonds present in lignin and hemicellulose were weaker for biomass after microwave NaCS treatment. The indicated peaks in the FTIR spectra were observed to be weaker when the hydrotrope concentration was higher during the pretreatment. Weaker peaks for biomass after NaCS treatment were observed at 1245 and 1320 cm^−1^ wavenumbers characteristic of C-O vibration in the syringyl ring in lignin, 1460 cm^−1^ corresponding to asymmetric bending in CH_3_ in lignin, 1520 cm^−1^ characteristic of the aromatic skeleton C-C stretching in lignin, and 1660 cm^−1^ corresponding to the acetate groups in hemicellulose [36,37,38]. FTIR analyses confirmed that lignins and hemicellulose content in the wheat stillage biomass samples decreased with increasing hydrotrope concentrations during microwave pretreatment.

In addition to the analysis of the composition and structure of the wheat stillage biomass after the microwave treatment with increasing NaCS concentrations (from 0 to 40% *w*/*v*), the suitability of the biomass as a raw material for the alcoholic fermentation process was also examined. The higher cellulose content in the biomass after pretreatment made it possible to obtain a higher concentration of glucose in the fermentation media due to enzymatic hydrolysis (Table 2).

The glucose concentration in the fermentation medium obtained from biomass after pretreatment with water (0% *w*/*v* NaCS) was 16.27 ± 0.18 g/L. However, when 40% *w*/*v* NaCS was used during pretreatment, glucose concentration increased by about 50%, to 24.40 ± 0.14 g/L (Table 2). This was a direct consequence of the fact that the cellulose percentage in the biomass after pretreatment with 40% *w*/*v* hydrotrope increased by about 59% as compared to the biomass after pretreatment without NaCS (Table 2). All samples of wheat stillage biomass obtained after pretreatment with increasing concentrations of NaCS (from 10 to 40% *w*/*v*) proved to be highly effective as a raw material for cellulosic bioethanol production. The media obtained from biomass pretreated with 10 or 20% *w*/*v* hydrotrope concentration were fully attenuated after 24 h of fermentation. The media pretreated with 40% *w*/*v* NaCS, with the highest glucose concentration, were fully attenuated after 48 h (Table 2). Importantly, no fermentation inhibitors, such as 5-HMF, furfural, 4-hydroxybenzoic acid, vanillin, syringaldehyde and trans-ferulinic acid, were found in the fermentation media. The use of microwave pretreatment with NaCS caused changes in the biomass structure of wheat stillage. The structural characteristic of lignin and hemicellulose were removed from the biomass of wheat stillage, which is confirmed by FTIR analysis. Wheat stillage biomass after microwave-assisted hydrotropic pretreatment is more susceptible to enzymatic hydrolysis as a result of changes in the structure of lignocellulose. The proposed method of wheat stillage pretreatment with the use of microwaves and NaCS enables the reduction of lignin content in biomass by 27% (Table 1). This indicates the intensification of the delignification process under the conditions of rising NaCS concentrations. Thus, the effectiveness of surface tension reducing substances in the process of removing hydrophobic substances, i.e., lignin, is confirmed. The obtained results clearly indicate that wheat stillage biomass is a more efficient raw material in second-generation ethanol production, if pretreated with NaCS. It was reported that the hydrolysates obtained using the maximum hydrotrope concentration in the microwave-assisted pretreatment showed no inhibitory effect on the fermentation activity of yeast. The improvement of efficiency in the production of bioethanol from biomass after hydrotropic pretreatment, although carried out without the use of microwaves, was also confirmed by other authors. Cotton stalk biomass after pretreatment with NaCS was a good raw material in ethanol production, providing 85.86% bioconversion efficiency to ethanol [39].

### 2.2. Selection of Cellulose Enzymatic Hydrolysis Parameters in the Biomass of Wheat Stillage after Microwave Treatment with 40% w/v NaCS

As the increase in the efficiency of cellulosic ethanol production from wheat stillage after microwave treatment with 40% *w*/*v* NaCS was confirmed, it was decided that the biomass prepared in this way would be used in further stages of the study. Optimization of cellulose enzymatic hydrolysis included the selection of the biomass concentration (125, 143, and 167 g/L), the dose of Cellic^®^ CTec2 (2.5, 5, and 10 FPU/g DW) and the duration of the process (24, 48 or 72 h) (Figure 3).

The highest efficiency of cellulose enzymatic hydrolysis, ca. 60%, was achieved after 72 h of the process, for a biomass concentration of 143 and 167 g/L and a dose of cellulolytic enzyme preparation of 10 FPU/g of biomass. When the enzyme preparation was used at a dose of 5 FPU/g of biomass, the hydrolysis efficiency only decreased by about 3% under identical process conditions (Figure 3). Using the lowest dose of cellulase complex (2.5 FPU/g of biomass) significantly decreased the hydrolysis efficiency by ca. 10–16% compared to the effectiveness obtained for higher doses of the preparation. Importantly, at this stage of the research, a slightly negative impact of higher biomass concentrations (after pretreatment with microwaves 40% *w*/*v* NaCS) on hydrolysis efficiency was observed. Despite the lower water activity in the solutions containing an increased biomass concentration (143 and 167 g/L), an increase in the hydrolysis efficiency (1–5%) was observed compared to the solutions containing 125 g of biomass per liter (Figure 3). The analysis of variance (ANOVA) showed a statistically significant influence of linear terms (biomass concentration, enzyme dose and hydrolysis time) on the effectiveness of the cellulose degradation process, with *p* < 0.05 (Appendix A). The correlation between enzyme dose and hydrolysis time was also determined at *p* < 0.05. The conducted analyses did not show any influence of the variable pairs: enzyme dose/biomass concentration and biomass concentration/hydrolysis time on the process efficiency. This showed that the biomass prepared with the use of a hydrotrope and microwaves could be very useful for the preparation of fermentation media. The presented results showed that the biomass obtained by the described pretreatment method was highly susceptible to hydrolysis with the use of a complex of cellulolytic enzymes. Other authors reported that enzymatic hydrolysis of cellulose after acid or alkali pretreatment proceeded with lower efficiency if a higher concentration of biomass in the medium was used. The efficiency of wheat straw biomass hydrolysis after acid or alkaline treatment was lower by about 50 to 60%, if the biomass concentration was increased from 2 to 20% DW [40]. The efficiency of enzymatic hydrolysis of corn stover after acid treatment (1% *w*/*v* H_2_SO_4_, 160 °C for 10 min) was lower by 15 to 50% at the biomass concentration of 20% than at the concentration of 1% [41]. The comparison of the results of this study with the results of other authors shows that the proposed method of microwave hydrotropic pretreatment of wheat stillage biomass is a promising technique for obtaining fermentation media that can increase the efficiency of cellulosic ethanol production.

### 2.3. The Course of Alcoholic Fermentation of Media Prepared from Wheat Stillage after Microwave Pretreatment with NaCS

The results obtained in the previous stage of the study showed the selection conditions for the preparation of fermentation media. The fermentation media were prepared in two versions of the biomass concentrations, 143 and 167 g/L, because the highest efficiency of cellulose hydrolysis was obtained at these concentrations. Cellic^®^ CTec2 preparation was used at a dose of 5 FPU/g DW of biomass. Moreover, a high degree of cellulose degradation into glucose was achieved at this dose. Viscozyme^®^ L was also added at a dose of 10 FBG/g DW of biomass to increase the concentration of fermentative sugars in the media. The preparation contains a complex of enzymes that hydrolyze β-1,3- and β-1,4-glycosidic bonds in hemicellulose (Table 3).

The use of an additional enzyme preparation that hydrolyzed hemicellulose increased the initial glucose concentration in the fermentation media by 3 to 3.5 g/L and galactose and xylose by 2.5 to 3 g/L (Figure 4). Enzymatic hydrolysis with the simultaneous action of cellulases and hemicellulases resulted in a similar concentration of glucose in the medium with a biomass content of 143 g/L (MHW2) as in the medium prepared with 167 g of biomass per liter but treated with Cellic^®^ CTec2 (MHW3 only) (Figure 4A). In all analyzed media, the complete attenuation took place after 48 h of the process, which indicated that the fermentation media contained nutrients guaranteeing a high level of metabolic activity of yeast (Figure 4A). As a result of the supporting action of hemicellulase, the galactose concentration in the medium increased. This sugar was absorbed by yeast mainly between 24 and 48 h of fermentation (Figure 4B).

The initial concentration of acetic acid was higher by ca. 0.1 g/L in fermentation media obtained after hydrolysis with enzymes degrading β-1,3- and β-1,4-glycosidic bonds compared to media in which only cellulose was hydrolyzed (Figure 5A). A higher initial concentration of acetic acid was due to the enzymatic degradation of hemicellulose and the release of acetate groups from its structure [3]. During fermentation, the concentration of acetic acid in the media increased throughout the entire process, while the concentration of glycerol only increased until the 48 h of fermentation (Figure 5).

A similar tendency was observed for the concentration of ethanol in the fermentation media, which was the effect of glucose bioconversion in the initial stage of fermentation (Figure 6A). After fermentation, the final ethanol concentration was highest in the media with simultaneous hydrolysis of cellulose and hemicellulose (MHW2—21.34 ± 0.38 g/L and MHW4—23.57 ± 0.10 g/L). It was, on average higher by ca. 3 g EtOH/L compared to the media prepared only with the Cellic^®^ CTec2 cellulase complex (Figure 6A). Also, the final fermentation yield was the highest (90–91%) in the media obtained with both enzyme preparations as compared to the theoretical one (Figure 6B). Fermentation inhibitors (such as 5-HMF, furfural, 4-hydroxybenzoic acid, vanillin, syringaldehyde or trans-ferulic acid), were not found in the obtained fermentation media. The present study clearly showed that the wheat stillage biomass after microwave pretreatment with 40% *w*/*v* NaCS was an effective raw material in producing cellulosic ethanol. The usefulness of wheat stillage as a raw material in cellulosic ethanol production was also confirmed in studies using mild-temperature diluted acid pretreatment.

Pretreatment of wheat stillage with sulfuric or phosphoric acid for 4.5 h resulted in the release of fermentable sugars, but the maximum ethanol concentration reported was low (ca. 13 g/L) [42]. Earlier studies also indicated the possibility of using the biomass of wheat stillage after hydrotropic treatment to produce cellulosic ethanol. However, in the previously proposed solution, the stillage biomass was first treated with a hydrotrope under elevated pressure, and then additionally, acid pretreatment was applied. The maximum concentration of ethanol in the fermentation medium obtained from stillage biomass prepared in this way was ca. 21.5 g/L, that is, it was lower by ca. 2 g/L compared to the results presented in this study [7]. It should be noted that, in this study not only a higher concentration of ethanol was obtained but also burdensome acid treatment was omitted. The elimination of harmful chemical reagents from the cellulosic ethanol production technology prevents the formation of an increased amount of by-products, including the fermentation inhibitors mentioned earlier [7].

## 3. Materials and Methods

### 3.1. Materials

The lignocellulosic raw material used in the study was biomass of wheat distillery stillage (Gospodarstwo Rolne, Radzicz, Poland) containing: cellulose 18.60 ± 0.30% DW, hemicellulose 34.10 ± 0.10% DW, and lignin 9.50 ± 0.20% DW. Before the experiments, the material was dried at 60 °C until constant weight, ground and sieved through a 1 mm sieve. Analytical reagents and HPLC grade solvents were supplied by Merck^®^ (Darmstadt, Germany). The standards used in the chromatographic analyses were HPLC grade, supplied by Sigma-Aldrich^®^ (St. Louis, Missouri, United States). Sodium cumene sulfonate (NaCS) (hydrotrope), in the form of a 40% (*w*/*v*) Stepanate SCS 40 preparation was supplied by the Stepan Company (Northfield, IL, USA).

### 3.2. Enzymatic Preparations

Cellic^®^ CTec2 preparation (Novozymes, Franklinton, NC, USA) containing a complex of cellulolytic enzymes (75 FPU/mL activity) was used in the enzymatic hydrolysis of cellulose. The preparation was applied according to the manufacturer’s instructions at pH 5.5 and 50 °C. Selected fermentation media were also prepared using Viscozyme^®^ L (Novozymes, Bagsvaerd, Denmark). This preparation exhibits endo-β-glucanase (xylanase, cellulase, hemicellulase) activity, 100 FBG g-1, and hydrolyses β-(1,3)- or β-(1,4)-glycosidic linkages in β-D-glucans. It was used according to the manufacturer’s instructions at a dose of 10 FGB/g DW at pH 5.5 and 50 °C.

### 3.3. Microorganisms

Active *Saccharomyces cerevisiae* preparation, strain Ethanol Red (Lesaffre Advanced Fermentations, Marcq-en-Baroeul, France) was used for fermentation. The yeast suspension (1.25 ± 0.12 10^9^ CFU/mL) was prepared by adding 1 g of yeast in 10 mL of 0.9% *v*/*v* NaCl at 30 °C (according to the manufacturer’s recommendations). The inoculum dose was 2 mL/L of fermentation medium, and the yeast cells viability was 94.3 ± 0.5%. The viability of the yeast cells (%) was estimated after staining with methylene blue and cell counting with Thom’s chamber [43].

### 3.4. Research Stages

#### 3.4.1. Assessment of the Effect of NaCS Concentration during the Microwave Pretreatment on the Biomass Composition of Wheat Stillage and the Course of the Fermentation Process

In the first stage of the research, the influence of NaCS concentration (0, 10, 20, and 40% *w*/*v*) during the microwave pretreatment under constant process conditions (microwave generator power 300 W, 30 min, 0.81 MPa, 170 °C, pH 8.0) was analyzed. The optimal process parameters (indicated in the previous sentence) were established in previous studies [12]. In further stages of the research, during the microwave pretreatment the NaCS concentration ensuring the highest concentration of fermentable sugars was used. The preparation of the raw material for the microwave pretreatment process at various NaCS concentrations was started by weighing about 2.5 g of stillage biomass and drying it at 130 °C to constant weight. The dried biomass (c.a. 2.3 g DW) was then transferred to HP-500 plus Teflon containers; 40 mL of 0 (demineralized water), 10, 20 or 40% *w*/*v* NaCS was added. The microwave pretreatment was performed using a Microwave Digestion System Mars 5 (CEM Corporation) under constant process conditions as described previously. After the pretreatment, the samples were filtered under reduced pressure, rinsed with 200 mL of demineralized water at 60 °C, and dried at 130 °C to constant weight. The pretreatment procedure involved ten replications and the biomass extractives were calculated based on the difference in sample weight before and after pretreatment. Biomass extractives (that is, nonstructural biomass components that can be removed by the extraction process) were measured as weight loss percentage. The parameter was calculated based on the difference in weight of lignocellulosic biomass before and after the pretreatment procedure [5]. The content of cellulose, hemicellulose and lignins in the obtained biomass was determined and the crystallinity index (CrI) was assessed using X-ray diffraction (XRD). The individual functional groups were identified and quantified using Fourier-transform infrared spectroscopy (FTIR).

Fermentation media were prepared from the biomass after pretreatment at a concentration of 100 g/L. For this purpose, 5 g of biomass were placed in 250 mL conical flasks, 45 mL of demineralized water was added, pH was adjusted to 5.5 with 1 M NaOH and 1 M H_2_SO_4_ and the adjusting volume to 50 mL (in triplicate). Then 335 µL (5 FPU/g DW) of Cellic^®^ CTec2 enzyme preparation was added to perform enzymatic hydrolysis of cellulose. Hydrolysis was carried out for 24 h at 50 °C with shaking at 80 rpm. In the solution obtained after enzymatic hydrolysis, the concentration of sugars and fermentation inhibitors were determined using HPLC. The sample was then inoculated with yeast milk (0.2 g DW of yeast per 1 L of cellulose medium). The flasks were capped with a fermentation tube filled with glycerin and incubated at 35 °C for 48 h. The composition of the fermentation media was examined by HPLC every 24 h of the process.

#### 3.4.2. Selection of the Enzymatic Hydrolysis Parameters of Cellulose

In the selection of parameters for cellulose enzymatic hydrolysis, the biomass of wheat stillage pretreated with microwaves under the process conditions described earlier with 40% *w*/*v* NaCS was used. Selection of the hydrolysis parameters was aimed at obtaining the highest possible concentration of glucose in the solution using the following variables: cellulolytic enzyme dose (2.5, 5, and 10 FPU/g DW), biomass concentration (125, 143, and 167 g/L) and hydrolysis time (24, 48, and 72 h). The hydrolysis procedure was started by weighing out 6.25, 7.15 or 8.35 g of wheat stillage biomass, adding 50 mL of demineralized water and adjusting pH to 5.5 with 1 M NaOH and 1 M H_2_SO_4_. Then 2.5, 5 or 10 FPU/g DW of Cellic^®^ CTec2 was added to the resulting solution. The solutions were then incubated at 50 °C for 72 h with shaking at 80 rpm. Three replications were made. The hydrolysis yield was calculated from the glucose concentration analyzed every 24 h according to [44].

#### 3.4.3. Assessment of the Use of Wheat Stillage after Hydrotropic Microwave Treatment in the Production of Bioethanol

Fermentation media were prepared from wheat stillage after microwave pretreatment in the earlier mentioned process conditions with 40% *w*/*v* NaCS. Two variants of fermentation media were prepared with the biomass concentration of 143 and 167 g/L. Two versions of enzyme preparations were used: one containing only Cellic^®^ CTec2 preparation with cellulase, the other with the aforementioned preparation, and additionally Viscozyme^®^ L containing the enzyme hydrolyzing β-(1,3)- or β-(1,4)-glycosidic bonds in β-D-glucans (Table 3). The purpose of testing various combinations of enzyme preparations was to obtain fermentation media with a high concentration of fermentable sugars. The procedure was started by weighing out 5 or 7.5 g (143 or 167 g/L) of pretreated biomass, adding 40 mL of demineralized water and adjusting pH to 5.5 with 1 M NaOH and 1 M H_2_SO_4_ and the adjusting volume to 45 mL. The enzyme preparations were then applied as shown in Table 3. The media were incubated at 50 °C with shaking at 80 rpm for 72 h. When the enzymatic hydrolysis process was completed, the temperature was lowered to 30 °C, and milk yeast was added to the media (0.2 g of yeast DW per liter). The flasks were capped with a fermentation tube filled with glycerin and incubated at 35 °C for 72 h. The composition of the fermentation media was analyzed immediately after inoculation and after 24, 48 and 72 h of fermentation. The fermentation yield after 72 h of the process was calculated based on the concentration of ethanol and sugars in the fermentation medium [45].

### 3.5. Analytical Methods

#### 3.5.1. Analysis of Lignocellulosic Biomass Components

The content of cellulose, hemicellulose and lignins in the biomass of wheat stillage was analyzed. The biomass components before and after pretreatment were determined using Fibertec^®^ 8000 according to [46].

#### 3.5.2. X-ray Diffraction (XRD) Analysis of Wheat Stillage Biomass

XRD analyses were performed on biomass after microwave pretreatment with various NaCS concentrations. Measurements were performed using a multipurpose X-ray diffractometer D8-Advance from Bruker operated in a continuous θ-θ scan in locked coupled mode with Cu-Kα radiation. The crystallinity index (CrI) was calculated based on the XRD analysis [47].

#### 3.5.3. FTIR Analysis of Wheat Stillage Biomass

FTIR analyses of biomass after microwave pretreatment with various NaCS concentrations were performed. The pretreated stillage samples were dried at 130 °C to constant weight, ground and screened through a 150 µm sieve. A 2 mg sample was mixed with 200 mg KBr (spectroscopic grade) and ground in an agate mortar for 5 min. Each sample was then subjected to 2 tons for 5 min in a hydraulic press (Specac Ltd. Kent, UK). The spectra were collected in the range 4000 to 400 cm^−1^ with a resolution of 4 cm^−1^ and 72 scans per sample on a Nicolet iS5 spectrometer (Thermo Fisher Scientific, Waltham, MA, USA).

#### 3.5.4. Analysis of Carbohydrates, Acetic Acid, Glycerol and Ethanol in Fermentation Media

Analysis of glucose, galactose, xylose, acetic acid, glycerol and ethanol in the fermentation media was performed using an HPLC model 1220 from Agilent Technologies^®^ (Palo Alto, CA, USA) equipped with a refractometric detector (HPLC-RID). The conditions for chromatographic separation have been described previously by [7].

#### 3.5.5. Determination of 5-HMF, Furfural and Phenolic Compounds in Fermentation Media

Phenolic compounds, that is, vanillin, 4-hydroxybenzoic acid, syringaldehyde, trans ferulic acid (lignin degradation products) as well as 5-HMF and furfural were also determined in fermentation media. The analysis was performed using the Agilent Technologies^®^ (Palo Alto, CA, USA) HPLC system, model 1260, equipped with a diode detector (HPLC-DAD). The conditions for chromatographic separation have been described previously by [7].

### 3.6. Statistical Methods

Statistical analysis was carried out using the TIBCO Software Inc. Statistica ver. 13 (Palo Alto, CA, USA). An ANOVA test and an HSD Tukey’s test were applied at the significance level of α < 0.05.

## 4. Conclusions

Microwave-assisted pretreatment with NaCS is an effective method of preparing wheat stillage biomass as a raw material for cellulosic ethanol production. The combined use of microwaves and a hydrotrope ensured a significantly reduced lignin content in the wheat stillage biomass by ca. 28% and hemicellulose content by about 79%. As a result, the conversion of sugars into ethanol was over 91%, and the ethanol concentration was over 23 g/L. The proposed technological solution creates the basis for integrating first and second-generation ethanol production technologies, through the use of waste biomass of processed starchy raw materials to produce cellulosic ethanol. The processing line integrating the 1st and 2nd generation technologies may, contain a common ethanol distillation stage.

## Figures and Tables

**Figure 1 molecules-27-06097-f001:**
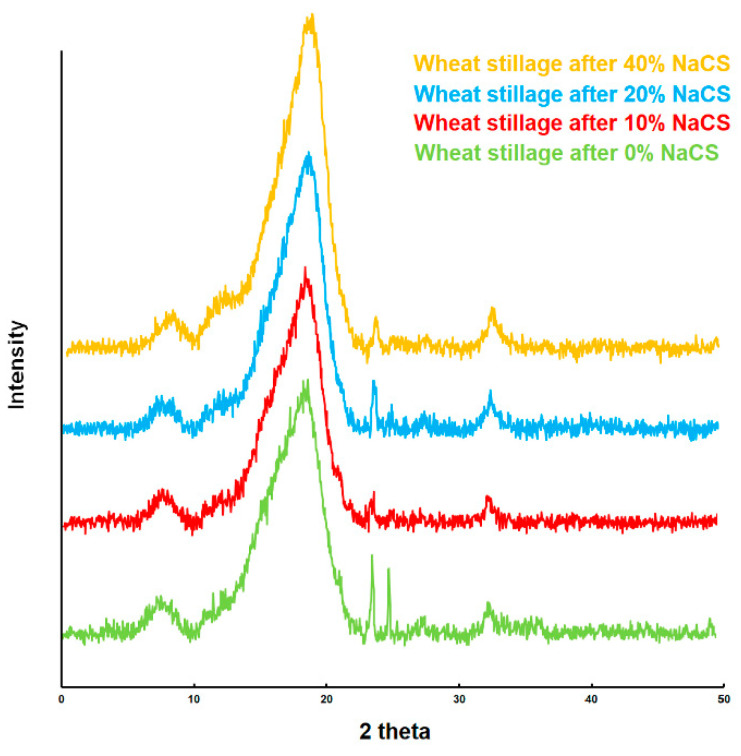
X-ray diffractograms of the wheat stillage samples after microwave-assisted hydrotropic pretreatment.

**Figure 2 molecules-27-06097-f002:**
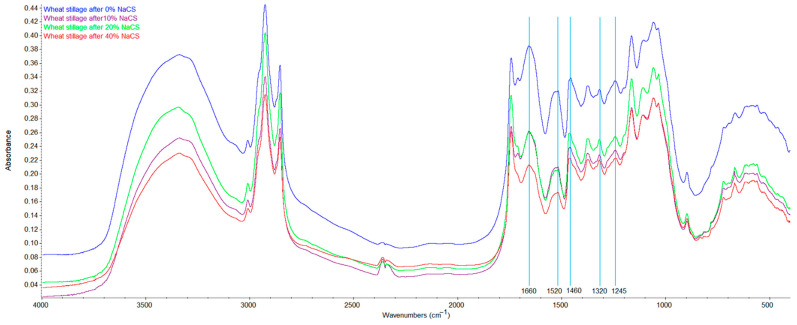
FTIR spectra of the wheat stillage samples after microwave-assisted hydrotropic pretreatment.

**Figure 3 molecules-27-06097-f003:**
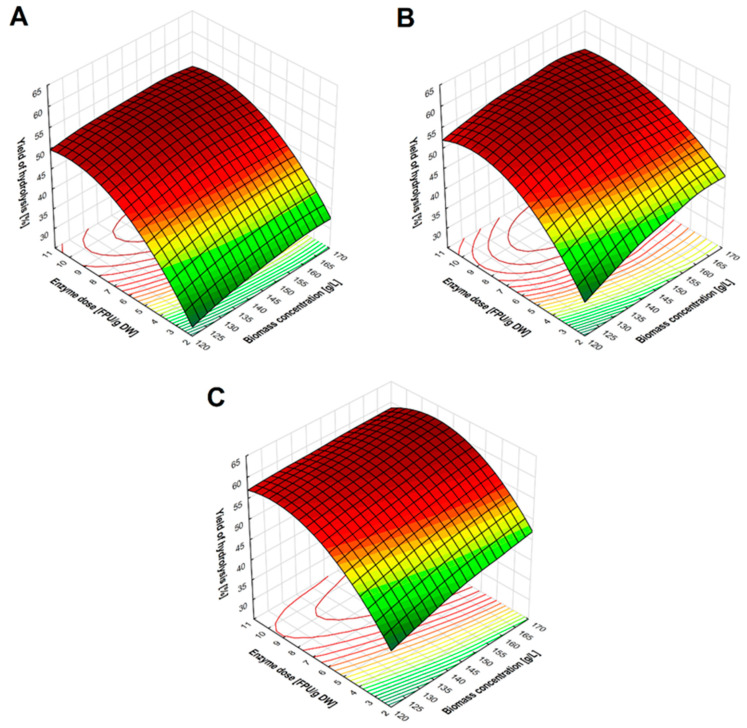
Graph showing the effect of biomass concentration and enzyme dose on the yield of cellulose enzymatic hydrolysis after 24 (**A**), 48 (**B**) and 72 (**C**) hours.

**Figure 4 molecules-27-06097-f004:**
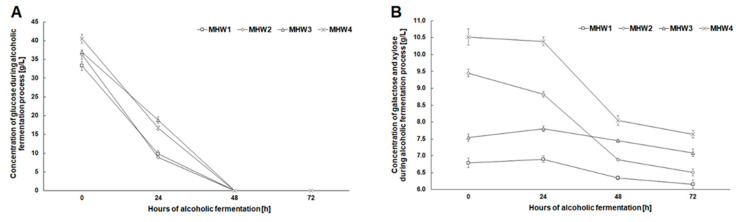
Changes in the concentration of glucose (**A**), and galactose and xylose (**B**) during alcoholic fermentation for wheat stillage concentrations 143 and 167 g/L after microwave-assisted pretreatment with NaCS.

**Figure 5 molecules-27-06097-f005:**
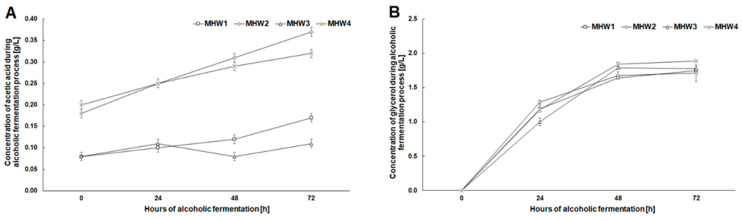
Changes in the concentration of acetic acid (**A**) and glycerol (**B**) during alcoholic fermentation for wheat stillage concentrations 143 and 167 g/L after microwave-assisted pretreatment with NaCS.

**Figure 6 molecules-27-06097-f006:**
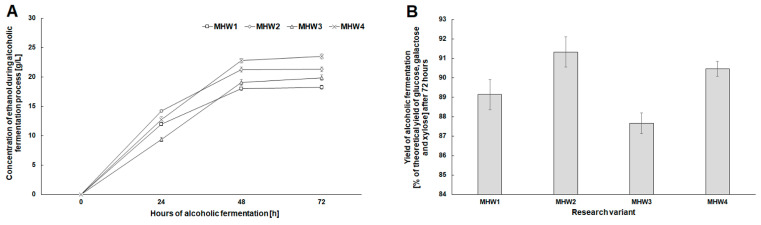
Changes in the concentration of ethanol during alcoholic fermentation (**A**), and alcoholic fermentation yield relative to the theoretical one after 72 h of the process (**B**) for pretreated wheat stillage concentrations of 143 and 167 g/L.

**Table 1 molecules-27-06097-t001:** Characteristics of wheat stillage after microwave-assisted pretreatment with NaCS.

Research Variant	Biomass Extractives [%]	Biomass Composition [% DW]			Crystallinity Index [%]
		Cellulose	Hemicellulose	Lignin	
Wheat stillage after microwave-assisted pretreatment in 0% NaCS	48.57 ±1.15 a	27.01 ± 1.26 a (NA)	8.05 ± 0.13 a (NA)	36.99 ± 0.13 a (NA)	70.15
Wheat stillage after microwave-assisted pretreatment in 10% NaCS	60.79 ±0.63 b	31.97 ± 0.70 b (+18%)	3.13 ± 1.10 b (−61%)	34.47 ± 0.10 b (−7%)	75.44
Wheat stillage after microwave-assisted pretreatment in 20% NaCS	64.76 ±1.02 c	36.51 ± 0.55 c (+35%)	2.46 ± 0.23 b (−69%)	29.72 ± 0.75 c (−20%)	78.38
Wheat stillage after microwave-assisted pretreatment in 40% NaCS	71.15 ±0.80 d	42.91 ± 0.10 d (+59%)	1.70 ± 0.40 b (−79%)	26.95 ± 0.40 d (−27%)	78.80

The mean values given in columns with different letter index are significantly different (α < 0.05). Percentage value in parentheses—change in biomass content in relation to biomass with the use of water (0% NaCS) during the microwave pretreatment. DW—dry weight. NA—not analyzed.

**Table 2 molecules-27-06097-t002:** Composition of the fermentation medium during subsequent hours of the fermentation process using wheat stillage in 100 g/L concentration after microwave-assisted pretreatment with NaCS.

Substrate for Fermentation Medium Preparation	Concentration [g/L] of the Fermentation Medium Components during Subsequent Hours of the Fermentation Process											
	0 h				24 h				48 h			
	Glucose	Galactose + Xylose	Glycerol	Ethanol	Glucose	Galactose + Xylose	Glycerol	Ethanol	Glucose	Galactose + Xylose	Glycerol	Ethanol
Wheat stillage after microwave-assisted 0% NaCS pretreatment	16.27 a ±0.18	4.06 a ±0.03	nd	nd	0.00 a ±0.00	3.21 a ±0.04	0.87 a ±0.03	7.80 a ±0.11	0.00 a ±0.00	3.03 a ±0.08	0.86 a ±0.02	7.67 a ±0.08
Wheat stillage after microwave-assisted 10% NaCS pretreatment	19.43 b ±0.22	3.07 b ±0.04	nd	nd	0.00 a ±0.00	2.34 b ±0.04	1.08 bc ±0.01	9.29 b ±0.11	0.00 a ±0.00	2.17 b ±0.01	1.09 b ±0.01	9.26 b ±0.14
Wheat stillage after microwave-assisted 20% NaCS pretreatment	20.79 c ±0.27	3.68 c ±0.07	nd	nd	0.00 a ±0.00	3.05 a ±0.04	1.12 b ±0.01	10.13 c ±0.16	0.00 a ±0.00	2.86 c ±0.04	1.13 b ±0.01	10.12 c ±0.12
Wheat stillage after microwave-assisted 40% NaCS pretreatment	24.40 d ±0.14	4.54 d ±0.04	nd	nd	3.26 b ±0.09	4.42 c ±0.03	1.03 c ±0.01	10.33 c ±0.10	0.00 a ±0.00	4.03 d ±0.02	1.14 b ±0.01	11.82 d ±0.18

The mean values given in columns with different letter index are significantly different (α < 0.05). nd—not detected.

**Table 3 molecules-27-06097-t003:** Characteristics of experimental variants.

Research Variant	Biomass Concentration of Wheat Stillage after Microwave-Assisted Pretreatment in 40% NaCS in Fermentation Medium [g/L]	Solvent	pH	Enzyme Preparations
MHW1	143	water	5.5	Cellic^®^ CTec2; 5 FPU/g DW
MHW2	143			Cellic^®^ CTec2; 5 FPU/g DW Viscozyme^®^ L; 10 FBG/g DW
MHW3	167			Cellic^®^ CTec2; 5 FPU/g DW
MHW4	167			Cellic^®^ CTec2; 5 FPU/g DW Viscozyme^®^ L; 10 FBG/g DW

DW—dry weight; FPU—filter- paper units; FBG—fungal β-glucanase units.

## Data Availability

Not applicable.

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
