# Peer review of "Cellulosic Ethanol Production Using Waste Wheat Stillage after Microwave-Assisted Hydrotropic Pretreatment"

_molecules, 2022, doi:10.3390/molecules27186097_

Round 1
Reviewer 1 Report
The research work of manuscript has certain novelty. In this paper, the raw materials and products of the reaction are analyzed, and the experimental results are discussed. However, I think that the analysis and discussion of the effect of microwave pretreatment on wheat distiller's grains can be improved in this paper. In addition, I also suggest the authors to further analyze and discuss the related process mechanism. It is suggested that the paper would be further revised and improved.
Author Response
We would like to thank the Reviewers for their comments and suggestions. We have revised the manuscript point by point according to the Reviewer’s comments. All the suggested changes are marked in yellow in the revised text and described below.
We hope that the quality and readability of our manuscript has been improved.
Response to Reviewer No. 1 comments:
The research work of manuscript has certain novelty. In this paper, the raw materials and products of the reaction are analyzed, and the experimental results are discussed. However, I think that the analysis and discussion of the effect of microwave pretreatment on wheat distiller's grains can be improved in this paper. In addition, I also suggest the authors to further analyze and discuss the related process mechanism. It is suggested that the paper would be further revised and improved.
As suggested by the Reviewer, the analysis and discussion of the impact of microwave pretreatment of wheat stillage biomass has been revised (in lines 67-76, 197-208).
Reviewer 2 Report
Grzegorz et al. evaluated the effect of microwave-assisted pretreatment of wheat straw with NaCS for bioethanol production. The pressure of the system during the reaction was rapidly increased by the action of microwaves, and the short pretreatment time resulted in less production of fermentation inhibitors. This work provides a reference for the development of biomass ethanol technology. However, before publication in this journal, I think the following modifications are required.
1. What is biomass extractives? Their composition, as well as their effect on the fermentation process, needs to be pointed out.
2. The components content, that is, the total content of cellulose, hemicellulose and lignin is about 70%, which is somewhat low.
3. In Table 1, what does [%DW] mean? If abbreviations are required, please give the full name. Also, there is a letter (a, b, c and d) after each data, I don't think they need to exist.
4. If a set of calculations for cellulose recovery, hemicellulose removal, and lignin removal can be added, it will better illustrate the effect of the pretreatment process.
5. The APPLICATION OF MICROWAVE IN THE field of pretreatment should be reviewed in the INTRODUCTION section. A recent study shows the benefits of microwave. Please refer to the related discussion. Tailored one-pot lignocellulose fractionation to maximize biorefinery toward controllable producing lignin nanoparticles and facilitating enzymatic hydrolysis,Chemical Engineering Journal,2022,450,138315.
Author Response
We would like to thank the Reviewers for their comments and suggestions. We have revised the manuscript point by point according to the Reviewer’s comments. All the suggested changes are marked in yellow in the revised text and described below.
We hope that the quality and readability of our manuscript has been improved.
Response to Reviewer No. 2 comments:
Grzegorz et al. evaluated the effect of microwave-assisted pretreatment of wheat straw with NaCS for bioethanol production. The pressure of the system during the reaction was rapidly increased by the action of microwaves, and the short pretreatment time resulted in less production of fermentation inhibitors. This work provides a reference for the development of biomass ethanol technology. However, before publication in this journal, I think the following modifications are required.
- What is biomass extractives? Their composition, as well as their effect on the fermentation process, needs to be pointed out.
The definition of biomass extractives has been given in the text (in lines 131, 434-437). It is the percentage of the substance extracted during the pretreatment (cf. Devendra, L.P.; Pandey, A. Hydrotropic pretreatment on rice straw for bioethanol production. Renew. Energ. 2016, 98, 2-8.). Biomass extractives are primarily water-soluble components of the stillage biomass, i.e. glycerol, waste yeast biomass and NaCS-soluble substances such as lignins and other hydrophobic substances.
- The components content, that is, the total content of cellulose, hemicellulose and lignin is about 70%, which is somewhat low.
As a result of microwave pretreatment with a hydrotrope, some biomass components such as lignins and hemicellulose are extracted. However, components that are poorly soluble in aqueous solutions (such as fats, residual starch, phenolic substances, Maillard reaction products) can still remain in the biomass. (Wilkie, A. C., Riedesel, K. J., Owens, J. M., 2000. Stillage characterization and anaerobic treatment of ethanol stillage from conventional and cellulosic feedstocks. Biomass and Bioenergy 19, 63-102.). Correction in lines 176-179.
- In Table 1, what does [%DW] mean? If abbreviations are required, please give the full name. Also, there is a letter (a, b, c and d) after each data, I don't think they need to exist.
As suggested by the Reviewer, an explanation of the abbreviation "DW" has been added below the table (in lines 137-139). The letter markings in the table have not been removed because they illustrate the results of the statistical analysis using the post-hoc test.
- If a set of calculations for cellulose recovery, hemicellulose removal, and lignin removal can be added, it will better illustrate the effect of the pretreatment process.
Table 1 now presents additional calculations illustrating the change in the lignocellulosic biomass content after microwave hydrotropic treatment in relation to the biomass composition after treatment with water.
- The APPLICATION OF MICROWAVE IN THE field of pretreatment should be reviewed in the INTRODUCTION section. A recent study shows the benefits of microwave. Please refer to the related discussion. Tailored one-pot lignocellulose fractionation to maximize biorefinery toward controllable producing lignin nanoparticles and facilitating enzymatic hydrolysis,Chemical Engineering Journal,2022,450,138315.
The Introduction section has been expanded to include the content indicated by the Reviewer. Correction in lines 67-76.
Reviewer 3 Report
The paper described an effective microwave assisted pretreatment. A few changes are needed before publication.
1. Page 4, lines 143-145. Please change the ‘content’ in to ‘percentage’. It is more accurate. Please do the same in other places in the manuscript.
2. Page 8, Figure 3. Add actual data points in the response surface plots. So we can see how close is the model to experimental data. Please add ANOVA table in the supplementary information and give simple description of ANOVA table results.
3. Lignin was removed in to the pretreatment media. Did the author find any phenolic compounds in the pretreatment media?
The author can cite following papers that have reported microwave pre-treatment of biomass
e.g.
Zhu, Z., Liu, Y., Gómez D.L., Wei T., Yang X., Simister R, McQueen-Mason S.J., Macquarrie D.J., Thermochemical pretreatments of maize stem for sugar recovery: Comparative evaluation of microwave and conventional heating, Industrial Crops and Products, 2021, 160, 113106.
Author Response
We would like to thank the Reviewers for their comments and suggestions. We have revised the manuscript point by point according to the Reviewer’s comments. All the suggested changes are marked in yellow in the revised text and described below.
We hope that the quality and readability of our manuscript has been improved.
Response to Reviewer No. 3 comments:
The paper described an effective microwave assisted pretreatment. A few changes are needed before publication.
- Page 4, lines 143-145. Please change the ‘content’ in to ‘percentage’. It is more accurate. Please do the same in other places in the manuscript.
As suggested by the Reviewer, the discussion of the results has been corrected (in lines 159, 160, 163, 164, 166, 167, 171, 172, 17, 185).
- Page 8, Figure 3. Add actual data points in the response surface plots. So we can see how close is the model to experimental data. Please add ANOVA table in the supplementary information and give simple description of ANOVA table results.
A table with the results of the ANOVA analysis and a description of the statistical analysis have been added (in lines 295-301, as suggested by the Reviewer). The statistical analysis of the results included only post-hoc tests (HSD Tukey's tests) determining statistically significant differences at the significance level α <0.05 and the analysis of variance. Figure 3 shows the effect of cellulolytic enzyme dose and biomass concentration on the effectiveness of the cellulose hydrolysis process (in lines 281-282). Mathematical modeling was not used in the study because the Design Expert software was not available to the authors. However, at further stages of the research, it is planned to use the above-mentioned software to generate models illustrating the enzymatic hydrolysis process.
- Lignin was removed in to the pretreatment media. Did the author find any phenolic compounds in the pretreatment media?
Fermentation media obtained from stillage biomass after microwave hydrotropic treatment were analyzed by HPLC-DAD method. Phenolic substances, such as 4-hydroxybenzoic acid, vanillin, syringaldehyde or trans-ferulic acid, were not found (information in lines 261-263, 362-364).
The author can cite following papers that have reported microwave pretreatment of biomass e.g. Zhu, Z., Liu, Y., Gómez D.L., Wei T., Yang X., Simister R, McQueen-Mason S.J., Macquarrie D.J., Thermochemical pretreatments of maize stem for sugar recovery: Comparative evaluation of microwave and conventional heating, Industrial Crops and Products, 2021, 160, 113106.
As suggested by the Reviewer, information on the efficiency of microwave pretreatment has been added (in lines 67-76).
Round 2
Reviewer 1 Report
The authors have revised the manuscript and I think it would be accepted after polished in English.
Author Response
We would like to thank the Reviewers for their comments and suggestions. We hope that the quality and readability of our manuscript has been improved.
Response to Reviewer No. 1 comments:
The authors have revised the manuscript and I think it would be accepted after polished in English.
As suggested by the reviewer, the manuscript has been revised by a qualified language service. The linguistic correctness of the entire manuscript was confirmed by a certificate.
